# Male fertility thermal limits predict vulnerability to climate warming

Belinda van Heerwaarden ⬡ [1,2✉] & Carla M. Sgrò[1]

Forecasting which species/ecosystems are most vulnerable to climate warming is essential to guide conservation strategies to minimize extinction. Tropical/mid-latitude species are predicted to be most at risk as they live close to their upper critical thermal limits (CTLs). However, these assessments assume that upper CTL estimates, such as CTmax, are accurate predictors of vulnerability and ignore the potential for evolution to ameliorate temperature increases. Here, we use experimental evolution to assess extinction risk and adaptation in tropical and widespread *Drosophila* species. We find tropical species succumb to extinction before widespread species. Male fertility thermal limits, which are much lower than CTmax, are better predictors of species' current distributions and extinction in the laboratory. We find little evidence of adaptive responses to warming in any species. These results suggest that species are living closer to their upper thermal limits than currently presumed and evolution/plasticity are unlikely to rescue populations from extinction.

[1] School of Biological Sciences, Monash University, Clayton, VIC, Australia. [2] Present address: School of BioSciences, The University of Melbourne, Melbourne, VIC, Australia. ✉email: belinda.vanheerwaarden@unimelb.edu.au

Projected increases in average temperature, as well as the frequency and duration of extreme heat events and heat-waves, pose a major risk to species persistence and biodiversity[1,2]. Understanding and predicting how species will respond to climate change and identifying which species are most vulnerable will be paramount to successfully managing biodiversity. Trait-based approaches (which compare estimates of thermal tolerance or performance to current and future habitat temperatures to calculate species' thermal safely margins or warming tolerance) have been utilised to examine vulnerability across different latitudes and habitats[3–11]. These studies suggest that tropical and mid-latitude species—which make up the vast majority of the world's biodiversity—are predicted to be most vulnerable because they already experience maximum habitat temperatures close to their upper thermal limits[3–11]. However, these studies have predominately focused on upper critical thermal limits (CTLs), the temperature at which adults stop moving or die[4–7], assuming that acute heat tolerance will be important in determining species' vulnerability to future climate change. This is largely driven by data that suggest that upper CTLs are better predictors of current distributions than other fitness measures, like optimum performance temperature ($T$opt)[3,6,12,13]. Nonetheless, associations between upper CTLs and species' distributions or abundance are often weak or absent[6,12,14,15], questioning the capacity for CTLs to predict future distributions under climate change. Significantly, emerging data suggest that upper CTLs—which are typically estimated on adults—may underestimate species' climate vulnerability[16–20]. Data from a recent comparative study on fish found that spawning adults and embryos have narrower thermal tolerance ranges than non-reproductive active adults or larvae[19]. There is also growing evidence suggesting that ectotherms, especially male ectotherms, may become sterile at temperatures much below their lethal (critical) thermal limits[16–18,21]. These studies suggest that upper CTLs using adults may underestimate vulnerability, yet explicit tests of whether other measures of thermal tolerance—such as fertility thermal limits (FTLs)—are better predictors of vulnerability to climate change are currently lacking.

Trait-based approaches to assessing vulnerability also largely ignore the extent to which adaptive responses (genetic evolution and phenotypic plasticity) may ameliorate the negative impacts of climate change[22–24]. This is despite the fact that adaptive responses may play a crucial role in species persistence and evolutionary rescue[25–28] and an increasing number of studies show that evolution and plasticity contribute to recent climate change responses in the wild (reviewed in[25]). Recent studies suggest that species, particularly those in the tropics, may have a limited capacity to increase upper CTLs via evolutionary or plastic shifts[6,29–37]. Yet, most studies on adaptive responses to increased temperatures have been performed under environmental conditions that bear little resemblance to conditions encountered by natural populations (but see[38]). This is significant because additive genetic variance (the genetic variance that underpins heritability and adaptive capacity) may change under different environmental conditions[36,39–44]. Whether adaptive evolutionary responses to climate warming, using ecologically relevant temperature fluctuations, are limited across tropical and widespread species is unclear. Furthermore, the extent to which other measures of thermal tolerance—such as male FTLs, which may underpin vulnerability to climate change—are able to evolve in response to climate warming is unknown.

Here we subject three tropical and three widespread species of Drosophila (Supplementary Table S1) to experimental warming using diurnally fluctuating temperatures reflecting current habitat temperature to investigate whether tropical and widespread species differ in their extinction risk when allowing for evolution and plasticity to contribute to responses. We find that tropical species are more vulnerable to experimental warming in the laboratory, becoming extinct at temperatures 1.6 °C on average lower than the widespread species. Using these extinction data, we then ask which thermal traits are better predictors of extinction risk, and expanding our analysis to ten species, assess and compare how close species' male upper FTLs and upper CTLs (critical thermal maxima (CTmax)) are to current habitat temperatures. We show that male upper FTLs are better predictors of extinction vulnerability and species distributions than CTmax. Importantly, we reveal for the first time that male upper FTLs have limited capacity to evolve or shift via plasticity. Our results suggest that while CTLs may predict warming vulnerability at a broad geographical scale, thermal safety margins/warming tolerance based on CTmax underestimate extinction risk.

## Results and discussion

**Tropical species go extinct at lower warming temperatures.** Using a fluctuating temperature regime (Supplementary Table S2) reflecting temperatures currently encountered during summer in tropical Australia (a region where all species are known to occur, https://www.taxodros.uzh.ch/), and increasing the average temperature by 0.2 °C every 2 weeks (roughly equivalent to every generation, Supplementary Table S1), we explored the effect of experimental warming on population persistence (Methods). We found that experimental warming triggered extinction in all species, but the temperature at which extinction occurred differed significantly across species (glm: $x^2 = 1108.0$, d.f. = 4, $P < 0.001$) —ranging from a temperature regime averaging 27.65 °C (minimum: 25.15 °C; maximum: 30.65 °C) in the tropical species D. sulfurigaster, to a regime averaging 30.7 °C (minimum: 28.2 °C; maximum: 33.7 °C) in the widespread cactophilic species D. buzzatii (Fig. 1, Supplementary Fig. S1, and Supplementary Table S3). All three tropical species succumbed to extinction at lower average fluctuating temperatures than the widespread species (Fig. 1 and Supplementary Table S2), and overall, the average extinction temperature for the tropical species (28 °C) was significantly lower than for the widespread species (29.6 °C) (glm: $x^2 = 1881.8$, d.f. = 1, $P < 0.001$). Thus, using ecologically relevant fluctuating warming temperature regimes—allowing for evolution and plasticity to contribute to responses—we show that tropical species are more at risk of extinction under equivalent levels of warming than the widespread species.

In addition to exploring differences in extinction, we also assessed whether species differed in the temperature at which they started to decline (when the census population size was significantly lower in the selected lines than the control lines) and when they stopped reproducing/became sterile (scored when no pupae were visible in cages). On average, species started declining 0.4 °C before they went extinct, while they stopped reproducing 0.32 °C prior to going extinct (Fig. 1 and Supplementary Table S3). Tropical species also starting declining and stopped reproducing at lower temperatures than the widespread species (Fig. 1 and Supplementary Table S3), suggesting that these tropical species may have lower FTLs than the widespread species.

**Male upper fertility thermal limits predict warming extinction.** CTLs in adults and $T$opt are often used to calculate extinction risk across latitude[3–7], but whether these traits are good predictors of species' vulnerability to climate change is not clear. Emerging evidence suggests that thermal tolerance may differ across life-stages[45–48] and upper FTLs (i.e., the temperature at which females or males become sterile) may be lower than CTLs[16,17,21]. To explore the extent to which different thermal traits vary within

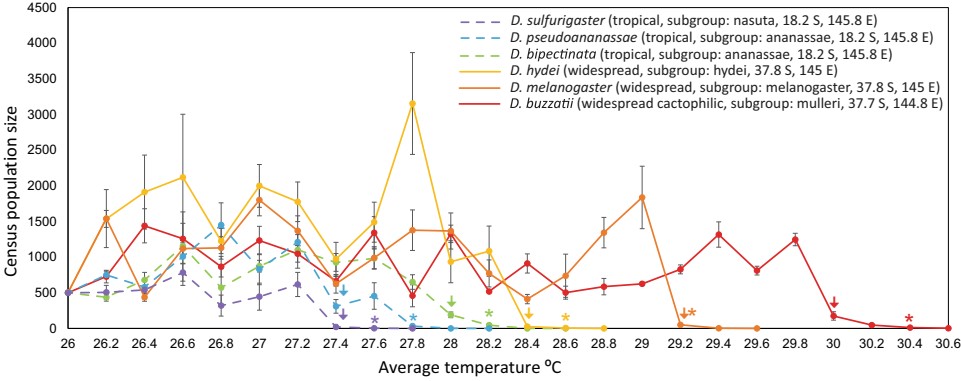

**Fig. 1 Changes in population size and extinction in response to gradual warming in tropical (dashed lines) and widespread (solid lines) *Drosophila* species.** Four replicate selection lines of *D. bipectinata* (green), *D. buzzatii* (red), *D. hydei* (yellow), *D. melanogaster* (orange), *D. pseudoananassae* (blue) and *D. sulfurigaster* (purple) were initiated under a fluctuating temperature regime averaging 26 °C (±3 °C) and exposed to a 0.2 °C increase in average temperature every 2 weeks (approximately one generation). Census population size (±SEM) was estimated at the end of each 2-week warming period by assessing the census population size of each replicate line. Information on species' distributions, subgroup and collection locations is the key. Coloured arrows indicate when selected lines were significantly lower (*P* < 0.05) than control populations (two-sided, independent samples *t*-test), and stars indicate when selected lines stopped reproducing/became sterile (scored when pupae were no longer present). Source data are provided as a Source Data file.

and across species, and assess which traits are better at predicting extinction vulnerability, we first estimated upper CTLs using knockdown CTmax estimated from standard ramping assays[6,35]. We then calculated developmental male and female upper FTLs and pre-adult (egg-to-adult) developmental upper viability thermal limits (VTLs) for each species by exposing them to fluctuating thermal regimes during development ranging from 25 to 34 °C (Supplementary Table S2 and Supplementary Fig. S2) and extracting $LT_{50}$ (male/female $FTL_{50}$ and $VTL_{50}$) and $LT_{80}$ (male/female $FTL_{80}$ and $VTL_{80}$) estimates (Methods, Supplementary Fig. S3). We also utilised published estimates of *T*opt (estimated using constant temperature regimes) for all species, except *D. buzzatii* where *T*opt estimates were not available[14].

CTmax and *T*opt both showed strong, significant positive linear associations with extinction temperature (Fig. 2), verifying that these traits were able to accurately predict which species were vulnerable to experimental warming. We also found strong and significant positive associations between extinction temperature and male $FTL_{50}$/$FTL_{80}$, female $FTL_{50}$/$FTL_{80}$ and $VTL_{80}$ (but not $VTL_{50}$), revealing that upper fertility and pre-adult viability thermal limits are also able to predict which species are more vulnerable to experimental warming (Fig. 2 and Supplementary Fig. S4). These thermal traits also showed significant positive associations with average temperature of decline and the average temperature they became sterile (Supplementary Fig. S4), suggesting that they are also good at predicting vulnerability prior to extinction.

Given that the species differed in their generation times (Supplementary Table S1), we also explored whether species with longer development times had lower extinction temperatures. We found no association between average extinction temperature and average development time at 25 °C (adjusted $R_2 = 0.27$, df = 5, $P = 0.16$) or 28 °C (adjusted $R_2 = 0.17$, df = 5, $P = 0.23$) (Supplementary Fig. S5). There was a high level of collinearity amongst all thermal traits, except *T*opt, which only showed a significant correlation with $VTL_{80}$ (Supplementary Table S4). Although other studies have found that different measures of thermal tolerance are largely independent[18,49–51], our results suggest that these traits may indeed be underpinned by similar physiology or genes, are under correlative selection or that they all individually contribute to extinction risk. Importantly, we have shown that all of these thermal traits are tightly associated with extinction

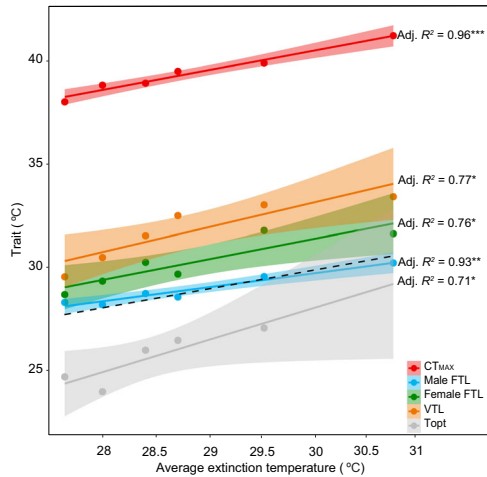

**Fig. 2 Associations between different estimates of thermal tolerance and average extinction temperature across six *Drosophila* species.** Critical thermal maximum (knockdown CTmax, red), male upper fertility thermal limits (male $FTL_{50}$, blue), female upper fertility thermal limits (female $FTL_{50}$, green), upper egg-to-adult viability thermal limits ($VTL_{80}$, orange) and optimum thermal temperature (*T*opt, grey) all show significant positive linear associations with average extinction temperature. Each point represents thermal tolerance for each species (see Supplementary Table S9 for values), the solid lines represent the fitted linear model and the shaded areas are the 95% confidence interval of this model. The dashed black line represents the direct relationship (slope = 1) between tolerance and average extinction temperature. Male FTLs are the only estimate of thermal tolerance that does not significantly differ from extinction temperature (two-sided paired *t*-test, *t* = 0.738, df = 5, *P* = 0.494).

vulnerability under increases in average temperature in the laboratory.

Although CTmax, *T*opt, $VTL_{80}$ and female/male $FTL_{50}$/$FTL_{80}$ were all able to predict which species were more vulnerable to warming, estimates of thermal tolerance using these measures differed: CTmax estimates in adults were on average much higher (8.5–10.4 °C) than all other measures of thermal tolerance, while $VLT_{50}$/$VTL_{80}$ estimates were higher than both male (2.0–2.6 °C higher) and female $FTL_{50}$/$FTL_{80}$ (0.7–1.1 °C higher) (Fig. 2 and Supplementary Fig. S4). In line with other studies suggesting that

male fertility in ectotherms may be more sensitive to temperature than female fertility[16,21,50], we found that male $FTL_{50}$ estimates were on average 1.3 °C lower than female $FTL_{50}$, indicating that male fertility underpins reproductive thermal limits in these species (Fig. 2 and Supplementary Figs. S3 and S4). Estimates of CTmax, $VTL_{50}/VTL_{80}$ and male/female $FTL_{50}/FTL_{80}$ were all higher than published estimates of $T$opt (Fig. 2 and Supplementary Fig. S4).

Because these traits differed in their sensitivity to thermal stress, we were interested in exploring the extent to which each trait was able to estimate individual extinction risk (i.e., which trait was closest to species' extinction temperatures). Male $FTL_{50}$ estimates were the closest to, and not significantly different from mean temperature at extinction or when they went sterile (Fig. 2, Supplementary Fig. S4, and Supplementary Table S5), illustrating that male FTLs are able to both accurately predict which species are most at risk, and how vulnerable they are to extinction. In contrast, all of the other thermal traits either over (CTmax, $VTL_{80}$ and female $FTL_{50}$) or under ($T$opt) estimated average extinction temperature (Fig. 2). Upper CTLs based on CTmax in adults were significantly higher than both the average and maximum temperatures at which species went extinct during experimental warming (averaging 10.6 °C above the average temperature at extinction and 7.6 °C above the maximum temperature at extinction), while $T$opt estimates were significantly lower (averaging 2.79 °C below the average temperature at extinction) (Fig. 2 and Supplementary Table S5). Estimated $VTL_{80}$ and female $FTL_{50}$ were also significantly higher than species' average extinction temperatures (Fig. 2 and Supplementary Table S5). These results suggest that populations are able to survive and reproduce 3.2 °C beyond their thermal optima ($T$opt), but are unable to persist under fluctuating temperatures with maximum temperatures that are 7.6 °C below their upper CTL (Fig. 2). In contrast, sublethal average temperatures that cause male sterility in species accurately predict both individual average extinction temperatures and overall extinction risk in the laboratory (Fig. 2).

**Male upper fertility limits shape current distributions**. These results reveal that male developmental upper FTLs are better predictors of extinction risk to increases in average temperature in the laboratory, but whether they influence current species' current distributions is not clear. To explore this, we extended our assessment of male $FTL_{50}$ estimates to include five tropical and five widespread *Drosophila* species (Methods) and then assessed the extent to which these traits associate with climatic variables across their distributional range. We also estimated CTmax in these ten species and used published estimates of $T$opt (which were only available for eight of these species[14]) to assess and compare the extent to which these different estimates of thermal tolerance/performance shape current distributions.

A study examining CTmax in almost 100 *Drosophila* species found only a weak association between CTmax and annual mean temperature (AMT) or maximum temperature of the warmest month ($T_{MAX}$), but detected a stronger association once precipitation ($P_{ANN}$) was included in the model (although it still did not explain much of the variation in CTmax ($R^2 = 0.29$))[6]. Across our ten species, we found a similar pattern: the best predictor model for CTmax (Table 1) included $T_{MAX}$ and precipitation in the wettest quarter (Pwet) ($R_2 = 0.39$, $P = 0.074$), where CTmax was lower in species from habitats with higher precipitation and temperatures (i.e., tropical species). The best predictor models for male $FTL_{50}$ also included $T_{MAX}$ and $P_{WET}$ (Table 1). However, $T_{MAX}$ and $P_{WET}$ were able to explain 79% of the variation in this trait (c.f. 39% for CTmax) (Table 1), indicating that male FTLs are better predictors of current species

**Table 1 Linear regression analysis was used to investigate the association between CTmax, male $FTL_{50}$ and $T$opt and climate across ten *Drosophila* species (only data from eight species were available for $T$opt).**

| Trait | Predictor | Adj. $R^2$ | Slope | $P$ | AIC |
|---|---|---|---|---|---|
| CTmax | Latitude (°) | 0.319 | 0.084 | 0.052 | 27.40 |
|  | AMT (°C) | 0.081 | −0.009 | 0.217 | 30.39 |
|  | $T_{MAX}$ (°C) | <0.001 | −0.014 | 0.471 | 31.72 |
|  | $T_{WARM}$ (°C) | 0.057 | −0.016 | 0.250 | 30.65 |
|  | $P_{ANN}$ (mm) | 0.301 | −0.011 | 0.058 | 27.66 |
|  | $P_{WET}$ (mm) | 0.289 | −0.002 | 0.063 | 27.82 |
|  | $T_{MAX} + T_{WARM}$ | 0.269 | – | 0.139 | 28.77 |
|  | $T_{MAX} + P_{ANN}$ | 0.366 | – | 0.084 | 27.35 |
|  | $T_{MAX} + P_{WET}$ | 0.390 | – | 0.074 | 26.97* |
|  | $T_{WARM} + P_{ANN}$ | 0.293 | – | 0.123 | 28.44 |
|  | $T_{WARM} + P_{WET}$ | 0.359 | – | 0.088 | 27.46 |
| Male $FTL_{50}$ | Latitude (°) | 0.278 | 0.114 | 0.067 | 35.10 |
|  | AMT (°C) | 0.196 | −0.017 | 0.112 | 36.10 |
|  | $T_{MAX}$ (°C) | <0.001 | −0.235 | 0.377 | 38.42 |
|  | $T_{WARM}$ (°C) | 0.136 | −0.027 | 0.159 | 36.83 |
|  | $P_{ANN}$ (mm) | 0.432 | −0.002 | **0.023** | 32.63 |
|  | $P_{WET}$ (mm) | 0.524 | −0.003 | **0.011** | 30.87 |
|  | $T_{MAX} + T_{WARM}$ | 0.476 | – | **0.043** | 32.49 |
|  | $T_{MAX} + P_{ANN}$ | 0.535 | – | **0.028** | 31.30 |
|  | $T_{MAX} + P_{WET}$ | 0.791 | – | **0.002** | 23.29* |
|  | $T_{WARM} + P_{ANN}$ | 0.438 | – | 0.055 | 33.18 |
|  | $T_{WARM} + P_{WET}$ | 0.790 | – | **0.002** | 23.36 |
| $T$opt | Latitude (°) | 0.282 | 0.082 | 0.101 | 24.08* |
|  | AMT (°C) | 0.028 | −0.009 | 0.315 | 26.50 |
|  | $T_{MAX}$ (°C) | <0.001 | −0.016 | 0.445 | 27.12 |
|  | $T_{WARM}$ (°C) | <0.001 | −0.012 | 0.452 | 27.15 |
|  | $P_{ANN}$ (mm) | 0.226 | −0.001 | 0.226 | 25.84 |
|  | $P_{wet}$ (mm) | <0.001 | −0.001 | 0.419 | 27.02 |
|  | $T_{MAX} + P_{ANN}$ | 0.001 | – | 0.430 | 27.27 |
|  | $T_{MAX} + P_{WET}$ | <0.001 | – | 0.739 | 30.00 |
|  | $T_{WARM} + P_{ANN}$ | 0.105 | – | 0.327 | 26.39 |
|  | $T_{WARM} + P_{WET}$ | <0.001 | – | 0.744 | 29.02 |

Climatic predictors included latitude (commonly used as a proxy for temperature), annual mean temperature (AMT), maximum temperature of the warmest month ($T_{MAX}$), mean temperature of the warmest quarter ($T_{WARM}$), annual precipitation ($P_{ANN}$) and precipitation of the wettest quarter ($P_{WET}$). The association between these traits and the predictor variables $T_{MAX}$, $T_{WARM}$, $P_{ANN}$ and $P_{WET}$ were also compared using a multiple regression approach. Significant $P$ values are highlighted in bold, while the model with the best Akaike Information Criterion (AIC) is highlighted with an asterisk. Source data are provided as a Source Data file.

distributions than CTmax. A model including $T_{WARM}$ and $P_{WET}$ was also able to explain 79% of the variation in male $FTL_{50}$ (Table 1), suggesting that both temperature extremes ($T_{MAX}$) and average temperatures during warmer months ($T_{WARM}$), in conjunction with precipitation, are important drivers of male developmental FTLs in these species. This finding contrasts with other studies showing upper thermal tolerance and fitness are linked to temperature extremes, rather than averages[6,8,12,52], but not unexpected since we measured FTLs by examining the effect of sublethal heat stress during development. In line with other studies on $T$opt (e.g.,[3,12]), we found no association between $T$opt and latitude or climate (Table 1), indicating that this trait is not useful for predicting current species distributions. Together, these findings suggest that male upper FTLs are better predictors of current distributions and are more closely aligned with the temperatures at which extinction occurs in the laboratory than CTmax or $T$opt.

**Upper critical thermal limits underestimate vulnerability**. A number of studies have examined biogeographic patterns of vulnerability by calculating the difference between a species' CTL or $T$opt and their current maximum (warming tolerance) or average (thermal safety margin) habitat temperatures[3–11]. However, given our results illustrating that male FTLs are lower than CTLs, and are better at predicting average extinction temperatures than either CTLs or $T$opt, these estimates may not accurately estimate individual risk. To explore this further, we used

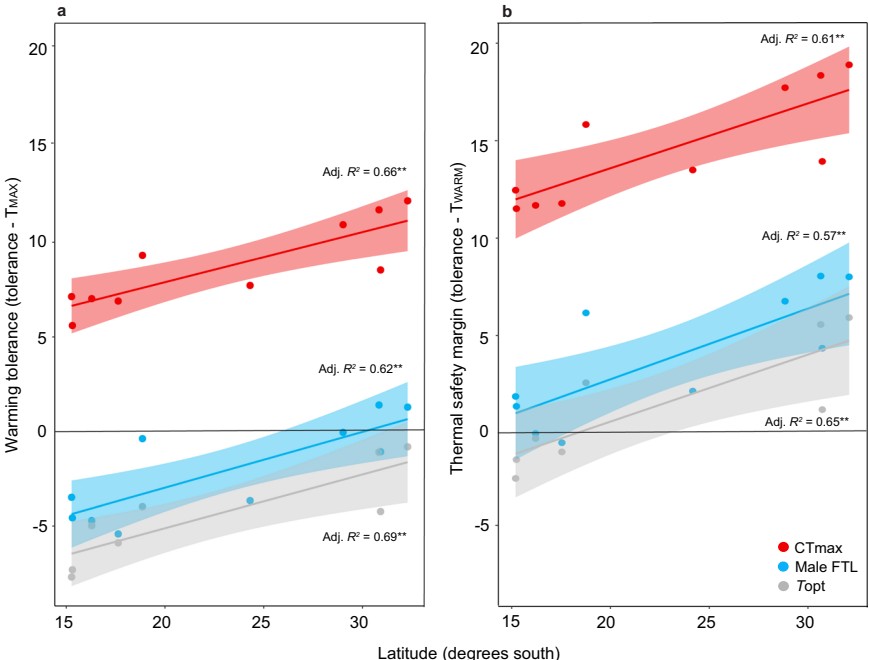

**Fig. 3 Thermal safety margins and warming tolerances across latitude for ten *Drosophila* species using different estimates of thermal tolerance. a** Warming tolerance (tolerance minus maximum habitat temperature ($T_{MAX}$)) and **b** thermal safety margins (tolerance minus average habitat temperature during summer months ($T_{WARM}$) calculated using critical thermal maxima (knockdown CTmax, red), male upper fertility thermal limits (male $FTL_{50}$, blue) and optimum thermal temperature (*T*opt, grey, note only eight species)) were lower in species from low latitudes. Warming tolerance (**a**) and thermal safety margins (**b**) using CTmax were higher than thermal safety margins using male FTLs (**b**). Each point represents warming tolerance/thermal safety margins for single species, the solid lines represent the fitted linear models and the shaded areas are the 95% confidence interval of these models. Source data are provided in the Source Data file.

these vulnerability measures to assess how close these ten tropical and widespread *Drosophila* species are to their upper male FTLs and assess the extent to which upper CLTs may underestimate vulnerability (Methods). The average thermal safety margin (which reflects how close species' thermal limits are to the average temperatures encountered during summer months) calculated using male $FTL_{50}$ was 3.8 °C (Fig. 3b). This was much lower than the average thermal safety margin calculated using CTmax (14.6 °C) (Fig. 3b). Since upper CTLs using CTmax estimate acute heat tolerance across hours of exposure, while we estimated male $FTL_{50}$ during development (i.e., days to weeks of exposure), warming tolerances—which estimate how close species' upper thermal limits are to the maximum habitat temperature—may be more appropriate for examining how close a species' upper CTL is to current habitat temperatures. The average warming tolerance for CTmax (8.7 °C, Fig. 3a) was still 4.9 °C higher than average thermal safety margins using male $FTL_{50}$ (Fig. 3b), suggesting that upper CTLs using CTmax may grossly underestimate climate risk.

Both warming tolerance and thermal safety margins for all traits were associated with latitude. Significantly, thermal safety margins using male $FTL_{50}$ were predominately less than two degrees in the tropical species (Fig. 3b), indicating that tropical species may already experience temperatures close to their male upper thermal fertility limits. In contrast, warming tolerances using CTmax were higher in these species (more than 5 °C, Fig. 3), suggesting that using CTmax to predict vulnerability underestimates individual climate change risk. To put this into the context of predicted climate changes, the thermal safety margin, using male $FTL_{50}$, of the tropically restricted species *D. bunnanda* is 0 °C, but its warming tolerance and thermal safety margin using CTmax are 11.7 and 7 °C, respectively

(Fig. 3). Thus, predicted increases in mean summer temperatures of 1–3 °C are likely to result in local extinctions of this species as adult males will be unable to reproduce despite the capacity to survive 7 °C increases in extreme temperatures above their current habitat temperatures.

Thermal safety margins using *T*opt were mostly below 0 for the tropical species, indicating that these species are already experiencing temperatures above their thermal optimum during warmer months. Other studies on squamate reptiles have also found that thermal safety margins using *T*opt are often below zero in tropical and sub-tropical species[5,8]. These findings—in conjunction with results showing this trait does not relate to habitat temperatures[3,6,12,13]—suggest that species may be using behavioural thermal regulation to evade temperatures above their *T*opt[10], or this trait is not important for driving population persistence. Given that species' decline/extinction temperatures were more than 2 °C above their *T*opt, and male FTLs were closely associated with both laboratory extinction and habitat temperatures, it is likely that male fertility, rather than *T*opt, is a key determinant of population persistence and vulnerability to high temperatures. Collectively, these findings illustrate that although CTmax and *T*opt may be useful for assessing geographical patterns in the distribution of climatic vulnerability, they are not good predictors of individual specie's climate change extinction risk.

Although we found that male $FTL_{50}$ estimates were strongly linked to extinction in the laboratory, they may potentially underestimate the effects of climate warming on population persistence in nature, as reductions in male fertility (sub-fertility) can occur prior to complete sterility and accumulate across generations[16]. We also did not consider that the impact of extreme temperatures, which in combination with above average

developmental temperatures, may exacerbate the effects of heat injury on fertility[17]. Future studies assessing both complete and sub-fertility within and across generations, under different thermal regimes, may further elucidate the potential impacts of climate warming on male fertility and population persistence.

**Male upper fertility thermal limits have little adaptive potential**. These results suggest that persistence under warming temperatures is underpinned by male developmental FTLs—which are lower than upper CTLs estimated using CTmax—and are precariously close to tropical species' current tolerance safety margins. Nonetheless, these safety margins reflect static measures and do not capture the capacity for species to increase their thermal tolerance via evolution or plasticity. To explore the potential for plasticity and evolution to increase upper thermal limits, we examined adaptive responses to experimental warming in species still producing enough offspring after 1.2, 2.2, 3.4 and 4.2 °C of warming, by comparing male fertility (number of offspring) and male sterility (proportion of fertile males) and CTmax in the control and selected lines developed under both control and selected developmental temperatures (Methods).

We found little evidence for adaptive evolutionary responses to warming in any species (Figs. 4 and 5 and Supplementary Tables S6–S8). Across both the tropical and the widespread species, we found no effect of selection on male sterility or fertility (Fig. 4 and Supplementary Tables S6 and S7). This is the first study to report no adaptive evolutionary responses in male upper FTLs to experimental warming. Consistent with previous studies showing little potential for evolutionary responses in CTmax in *Drosophila* and other insect species[37,38,53–56], we also found no effect of experimental warming on CTmax (Fig. 5 and Supplementary Table S8). These results suggest that both tropical and widespread species may have little potential to increase their upper thermal limits via adaptive evolution in response to either extreme temperatures[37,54–56], or extended heatwaves (increases in average temperature)[38,53].

We also found limited potential for adaptive developmental phenotypic plasticity to increase upper thermal limits. Although we detected a significant positive effect of developmental temperature on CTmax in *D. melanogaster* after 2.2 °C of warming, and in *D. buzzatii* after 3.4 °C of warming (Fig. 5 and Supplementary Table S8), similar to other studies examining plasticity in CTmax[34,35], increases were small and not adequate to compensate for increases in average temperature (*D. melanogaster*: 0.1 °C increase in CTmax for each degree of warming; *D. buzzatii*: 0.05 °C increase in CTmax for each degree of warming). A significant effect of developmental temperature was also observed for male fertility/sterility in *D. bipectinata*, *D. hydei*, *D. pseudoananassae* after 1.2 °C warming and *D. buzzatii* after 4.2 °C of warming (Fig. 4 and Supplementary Tables S6 and S7). However, these temperature effects were largely driven by a reduction in fertility/sterility under warmer selection developmental temperatures, especially as species approached extinction (Fig. 4). These results indicate that evolution and adaptive plasticity are contributing very little to responses to experimental warming, particularly for male fertility limits, which show no positive effects of developmental plasticity, nor signals of evolution.

Although it is unclear what is driving these patterns, several studies suggest that heritability may be low for CTmax[37,54] and male fertility at high temperatures[50], which will limit evolutionary responses. There is also evidence that plasticity for CTmax may be lower at higher developmental temperatures[35,57,58], It is also possible that transgenerational phenotypic plasticity (i.e., phenotypic effects lasting across generations) may impact responses to warming. Transgenerational plasticity has been highlighted as a

potential mechanism for mediating the impacts of climate change[59], but a recent study on male fertility in the flour beetle *Tribolium castaneum* found that heatwaves caused negative transgenerational impacts on male reproduction[16], suggesting that it may also be maladaptive for some traits. Because we reciprocally transplanted control and selected lines, we were able to assess whether transgenerational plastic effects may be influencing responses to warming here[60,61]. If negative transgenerational effects were hindering evolutionary responses, or limiting our ability to detect evolutionary responses in the selected lines, we expected that selected lines would do significantly worse than control lines under control and/or selected developmental temperatures[60,61].

We found no significant interactions between developmental temperature and selection treatment for CTmax or male sterility (Supplementary Tables S6 and S8 and Figs. 4 and 5), but a significant interaction between developmental temperature and selection treatment was detected for male fertility in *D. pseudoananassae*, *D. melanogaster* and *D. buzzatii* after 1.2 °C of warming (Supplementary Table S7). In all of these cases, the selection lines were not significantly lower than the control lines at either developmental temperature (Fig. 4), but there was a trend for lower fertility in the selected lines under control developmental temperatures for *D. pseudoananassae* and *D. melanogaster* after 1.2 °C of warming (Fig. 4), which is consistent with negative transgenerational effects of warmer selection temperatures.

It is also possible that extreme temperature events may be more effective in driving evolutionary responses in these traits, particularly for CTmax. However, other studies that have directly selected upon CTmax or acute heat knockdown have also failed to show a sustained response[49,55,56], suggesting that evolutionary responses to both increases in average and extreme temperatures may be limited. Alternatively, perhaps the experimental warming regime we used did not actually reflect ecologically realistic scenarios. By using a fluctuating temperature regime based on average summer air temperatures and increasing the temperature every generation (see Methods), the impacts of microclimate and seasonal temperature variation were not considered. Consequently, the level of warming may have been more rapid or intense than in nature, where exploitation of cooler microhabitats through behavioural thermoregulation, or recovery from heat injury during cooler months is possible[10,24]. This may be important for male fertility in particular, as past studies on temperature induced male sterility in *Drosophila* have shown that it is possible to recover fertility, depending on the intensity of the thermal stress[17,18]. Consequently, the extinction temperatures estimated in this study may differ from extinction temperatures in nature. Nonetheless, given that successive and extended heatwaves are projected in the coming decades[1,2], warmer than average temperatures across multiple generations (which may be particularly relevant to small, short-lived ectotherms), in conjunction with increased temperature extremes, are likely to push many species beyond their fertility limits and elevate extinction risk. Projected increases in night time temperatures may also further reduce opportunities for fertility recovery and exacerbate effects of high temperatures on fertility[62].

Understanding and forecasting species' responses to climate change is one of biology's greatest challenges. Although tolerance to extreme temperatures is thought to be a key driver of vulnerability to climate change[8,12,52,63,64], using a long-term warming experiment employing ecologically relevant temperature fluctuations, we show that the loss of male fertility at sublethal temperatures is a strong predictor of extinction temperature in the laboratory. Importantly, while we illustrate that current assessments of climate risk using CTLs can accurately assess the

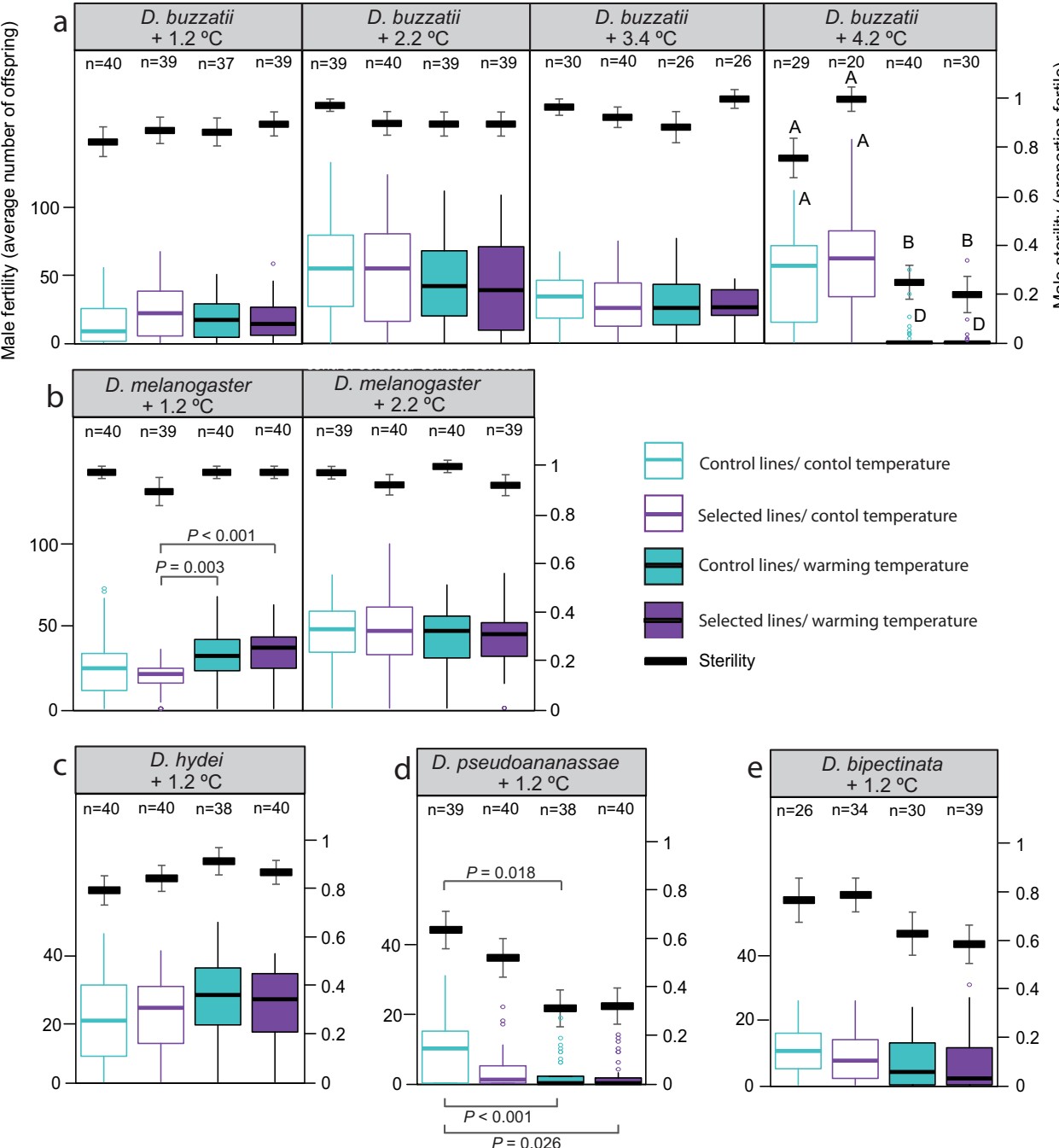

**Fig. 4 The response of male sterility and fertility to gradual experimental warming in five species of *Drosophila*.** Mean male sterility ± SEM (proportion of fertile males, solid black dash) and fertility (number of offspring, coloured boxplots denoting the median (solid horizontal lines), upper and lower quartiles (boxes), 1.5× interquartile range (vertical lines) and outliers (circles)) across four replicate control (aqua) and four replicate selected lines (purple) of **a** widespread *D. buzzatii* after 1.2, 2.2, 3.4 and 4.2 °C of warming; **b** widespread *D. melanogaster* after 1.2 and 2.2 °C of warming; **c** widespread *D. hydei* after 1.2 °C of warming; **d** tropical *D. pseudoananassae* after 1.2 °C of warming; and **e** tropical *D. bipectinata* after 1.2 °C of warming. Fertility/sterility was estimated by reciprocally transplanting eggs from control and selected lines to developmental temperatures reflecting both the control thermal regime (26 ± 3 °C, open boxes) and the warming thermal regime (solid boxes) at the generation of testing (e.g., 27.2 ± 3 °C after 1.2 °C of warming) and estimating male fertility/sterility at those developmental temperatures. Significant differences (two-sided Tukey post hoc test) between selection lines/developmental temperature within a generation of warming are shown by lines with corresponding *P* values or differing letters (**a** versus **b**: *P* < 0.01, **a** versus **d**: *P* < 0.001). *n* = total number of individuals per temperature/treatment. Source data are provided as a Source Data file.

geographical distribution of vulnerability, they are likely to be underestimating individual extinction vulnerability, particularly in tropical species which are already sitting close to their reproductive thermal limit. These findings, along with other

studies pointing to the heat sensitivity of male fertility in other organisms[16–18,21], highlight the importance of incorporating FTLs into estimates of climate change vulnerability. Critically, no strong signals of adaptive evolutionary or plastic responses to

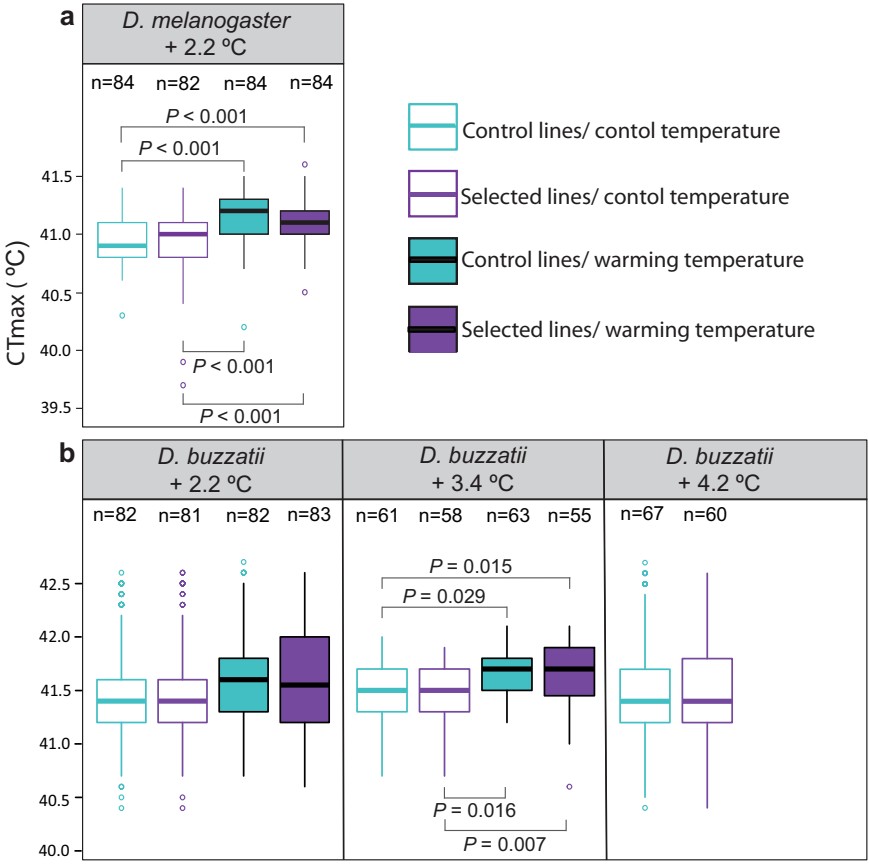

**Fig. 5 The response of critical thermal maxima (CTmax) to gradual experimental warming in two species of *Drosophila*.** Knockdown CTmax (boxplots denoting the median (solid horizontal lines), upper and lower quartiles (boxes), 1.5× interquartile range (vertical lines) and outliers (circles)) across four replicate control (aqua) and four replicate selected (purple) lines of **a** widespread *D. melanogaster* after 2.2 °C of warming, and **b** widespread *D. buzzatii* after 2.2, 3.4 and 4.2 °C of warming. CTmax was estimated by reciprocally transplanting control and selected lines to developmental temperatures reflecting both the control thermal regime (26 ± 3 °C, open bars) and the warming thermal regime (solid bars) at the generation of testing (e.g., 28.2 ± 3 °C after 2.2 °C of warming). Significant differences (two-sided Tukey post hoc test) between selection lines/developmental temperature within a generation of warming are shown. *n* = total number of individuals per temperature/treatment. Source data are provided as a Source Data file.

experimental warming, in either CTmax or male fertility at high temperatures, suggest that evolution or plasticity is unlikely to rescue species living close to their upper thermal limits from warming under current emission scenarios[2].

## Methods

**Sample collection.** The ten species of *Drosophila* (five tropical and five widespread species) were collected from sites around Kirrama or Melbourne, Australia (Supplementary Table S1) in April/May 2017. Field inseminated females of each species were used to establish iso-female lines, and one to two generations after field collections, mass-bred populations of all each species were made by combining 20 females and 20 males from 10 to 20 iso-female lines per species (Supplementary Table S1). Mass-bred populations were maintained as discrete generations, in large population sizes (>1000 individuals) across 6 × 250 ml bottles on potato dextrose agar medium at a constant 25 °C under a 12:12 light cycle.

**Initial thermal trait assessment.** After 6–15 generations of mass-breeding (Supplementary Table S1), upper viability and developmental fertility thermal limits in males and females were estimated by assessing egg-to-adult viability and male and female sterility at different fluctuating temperatures, averaging from 25 to 29 ± 3 °C (Supplementary Table S2 and Supplementary Fig. S2). This thermal regime was chosen as these fluctuations are similar to those encountered in summer tropical Queensland[43] (Bureau of Meteorology, http://www.bom.gov.au/climate), a location where all species can be found (https://www.taxodros.uzh.ch/). As the 29 °C fluctuating temperature regime was not warm enough to induce sterility/viability lethality in the more heat tolerant species (Supplementary Fig. S3), we repeated these assessments including additional temperatures of 30, 31, 32 and 33 in these species two generations later (Supplementary Fig. S2). *D. bunnanda*, *D.*

*birchii* and *D. serrata* were assessed in a different generation of culture than the other seven species.

*Viability thermal limits.* VTLs were assessed by estimating egg-to-adult viability at each fluctuating temperature regime (Supplementary Fig. S2 and Supplementary Table S2). Approximately 500 adult flies of each species, which had undergone development at a constant 25 °C, were placed in three population cages with a lid containing standard fly food (see above) stained with food dye and covered with a layer of live yeast to stimulate oviposition. Flies were allowed to lay eggs for approximately 12 h at 25 °C, after which time flies were removed. In the first round of assessments (see above), for each species, 20 eggs were then placed into 20 replicate vials per fluctuating temperature to develop. We also repeated this in the more tolerant species by picking 20 eggs into 10 replicate vials per fluctuating temperature from 28 up to 34 °C. The number of adults emerging from each vial was scored until no adults had emerged for 48 h. Adults emerging at 25 and 28 °C were scored at 8-h intervals so average development time could be calculated.

*Male fertility thermal limits.* Male sterility at different developmental temperatures was estimated by assessing the proportion of fertile males after developing at each fluctuating temperature (Supplementary Fig. S3). For each species, eggs were collected as described above, but for each species 40 eggs were placed into three replicate vials per temperature, as well as 15 vials of 40 eggs at 25 °C. Twenty virgin males were collected from each fluctuating developmental temperature, and 100 virgin females of each species were collected from 25 °C. At each fluctuating temperature regime, 19–20 (average 19.9) males were each paired with two virgin females in a single vial 4–5 days after emerging, and allowed to mate and lay eggs for four days. Both males and females were then discarded. Males were scored as fertile if larval activity was evident 7 days later. Vials with no visable larval action were double checked under a microscope.

*Female fertility thermal limits.* Female sterility at different developmental temperatures was estimated by assessing the proportion of fertile females after developing at each fluctuating temperature (Supplementary Fig. S3). For each species, eggs were collected as described for male FTLs above. Twenty virgin females were collected from each fluctuating developmental temperature, and 100 virgin males of each species were collected from 25 °C. At each fluctuating temperature regime, 16–20 (average 19.6) females of each species were each paired with two virgin males in a single vial 4–5 days after emerging, and allowed to mate and lay eggs for four days. Both males and females were then discarded. Females were scored as fertile if larval activity was evident seven days later.

*CTmax.* CTmax was assessed in 7-day-old adult female flies, which had developed at a constant 25 °C. This trait shows very little sexual dimorphism[6] and CTmax estimates are often measured on females. Individual flies were placed into 10-mL dry vials sealed and submerging in a water bath heated to 25 °C. The temperature was then gradually increased at a rate of 0.1 °C per minute. CTmax was scored as the temperature at which flies went into a heat coma (i.e., no movement)[6,35]. CTmax was assessed across two runs with two different scorers/observers in a randomised block design.

**Experimental evolution.** Experimental evolution was used to investigate whether tropical and widespread species show adaptive responses to warming using ecologically realistic temperature fluctuations, to explore whether species differ in the temperature at which they go extinct and assess which thermal traits are better at predicting extinction temperature. Three tropical (*D. sulfurigaster, D. bipectinata, D. pseudoananassae*) and three widespread *Drosophila* species (*D. melanogaster, D. buzzatii, D. hydei*) were selected, as they differ in their distribution, thermal tolerance and phylogenetic relationships[6] (Supplementary Table S1). Six to fourteen generations after field collection (Supplementary Table S1), eight replicate lines of 500 individuals (250 males and 250 females) were initiated for each species to establish four replicate control lines and four replicate selection lines per species.

The control lines were maintained in controlled temperature and humidity cabinets under a fluctuating temperature regime averaging 26 °C (ranging from 23.5 to 29 °C (Supplementary Table S2), constant 90% relative humidity, 12:12 light), while after 14 days at the control temperature regime (the time taken for all species to undergo at least one full generation (Supplementary Table S1)), the selected lines were subjected to experimental warming, involving increasing the average temperate by 0.2 °C every 14 days, henceforth referred to as a generation. We chose to use the average 26 °C temperature regime as the control conditions because an average temperature of 26 °C is close to their laboratory constant temperature of 25 °C, where they had been maintained since collection from the field (Supplementary Table S1) and it is close to the average temperature and temperature fluctuations commonly experienced during warmer months in tropical Queensland[43] (http://www.bom.gov.au/climate), where all of these species can be collected (https://www.taxodros.uzh.ch/). Egg-to-adult viability and the proportion of fertile males and females was also high in all six species at the control temperature regime (Supplementary Fig. S3).

While increases in mean temperature, as well as the frequency and duration of extreme heat events, are projected in coming decades[1,2], we chose to only increase the mean temperature during this experiment. Given that the current number of replicate cages was the maximum we could manage experimentally, we chose to thoroughly examine the effect of increases in mean temperatures (where future projections are more clearer than changes in variances/extremes[1,2] and maximise the replication of the number of species and replicate lines, rather than examine both increases in mean and extreme temperatures. Nonetheless, because we used fluctuating temperatures, the maximum temperature was also increased every generation so the effect of higher maximum temperatures was incorporated into our design. We chose not to relax lines (i.e., put selection lines back at control temperatures) between generations because lines were maintained as overlapping generations and species had differing developmental times, which made it difficult to choose a time point to relax.

Although this accelerated warming scenario is not entirely ecologically realistic as it does not encompass seasonal (where reprieve from hotter summer temperatures will occur during cooler months) or microclimate (which may allow species the opportunity to use behavioural thermoregulation to avoid thermal stress) temperature variation, as mentioned above, it is difficult to capture this complexity in the laboratory. Nonetheless, it does allow us to compare extinction risk and adaptation across species in response to equivalent levels of gradual warming in the laboratory. Furthermore, given that successive and extended heatwaves are projected in the coming decades[1,2], this warming scenario may become increasingly relevant to small, short-lived ectotherms like *Drosophila*.

Each of the replicate control and selected lines were maintained in separate plastic cages (length 188 mm, width 129 mm, height 174 mm, with air holes covered with stockings on the top (length 140 mm, width 140 mm), as overlapping generations, where a 250 ml plastic bottle containing 60 ml of potato dextrose agar medium was added every 3–4 days (every Monday and Thursday). A rotation system was used, whereby the oldest bottle was removed and replaced with a fresh bottle of food, so that only five bottles remained in one cage at any time, and each bottle remained in the cage for 14 days to allow a complete generation from egg to adult.

Every 14 days, the census size of each cage was assessed using a Drosophila Funnel Monitor (TriKinetics Inc). As it would be too laborious to count every adult fly in 48 replicate cages each generation, at generation 2, we placed an extra bottle with fresh food in each replicate control cage for 24 h to attract the majority of adult flies, and counted all flies in this bottle, as well as in and out of other bottles in the cage (i.e., all adult flies of each replicate line), to establish the proportions of flies in and out of the extra 'counting bottle' for each species. This proportion was then used to estimate total census size in each subsequent generation for each replicate line. Every 14 days, when the census size was being assessed, cages were cleaned and all adult flies were removed from each cage and transferred into a clean cage along with the remaining food bottles containing eggs, developing larvae and pupae, as well as the adults that were sensed in the 'counting bottle'. The 'counting bottle', which had freshly laid eggs, was then placed at 18 °C as a back-up if any replicate lines were lost due to unforeseen experimental issues. When flies were no longer present in the extra 'counting bottle', we checked for, and counted adult flies in all of the other bottles and out in the cage. We also checked for the presence of pupae in bottles, to assess fertility/reproductive capacity each generation. Lines were considered extinct when no adult flies or pupae were present.

**Assessing traits in experimental evolution lines.** After 1.2, 2.2, 3.4 and 4.2 °C of warming, we assessed knockdown CTmax and male fertility/sterility in the replicate control and selected lines. These time points were chosen because we were unable to assess such a large number of replicates every generation and to allow time for adaptation to occur. We reciprocally transplanted each line by placing two additional food bottles into each cage for 24 h to allow flies to lay eggs. We then removed these bottles and placed one bottle in a temperature/humidity cabinet programmed for control regime conditions and the other in a temperature/humidity cabinet with warming conditions reflecting the current level of warming (e.g., 1.2 °C warmer). Flies then were allowed to develop into adults. All replicate lines where enough offspring were collected from these bottles were assessed. Unfortunately, *D. sulfurigaster* was declining after 1.2 °C of warming and not enough offspring were collected to be assessed for evolution. After 3.4 °C of warming, one *D. buzzatii* control replicate line was not assessed at either temperature, and one selected line at the selected temperature, due to experimental error. One *D. buzzatii* selection line was not assessed after 4.2 °C of warming because no offspring emerged from either the selected or control temperature treatment due to a loss of fertility, while not enough offspring emerged from another selection line under the control temperature treatment. One of the *D. bipectinata* control replicates was lost after only a few generations of warming due to experimental error and was not assessed.

To assess male fertility/sterility, 6–10 males (average 9.6) were collected from each bottle from each temperature regime, then 4 days later, each male was mated with two virgin females, which had developed at constant 25 °C in single vials with 7 ml of fly media. They were then given 48 h to mate and lay eggs, and emerging offspring were counted 10–14 days later (depending on each species development time). Flies were mated and developing offspring were maintained at the same control or selected temperatures for the entire duration of the assessment.

To assess CTmax, we collected 13–21 females per replicate (average 18.7), per developmental treatment. We assed CTmax on 5–6-day-old adult females using standard ramping assays described above. Due to the large number of replicates, lines, temperatures and treatments needing to be assessed at 1.2 °C of warming, and the low likelihood that species would show changes in CTmax after this low level of warming, CTmax was regrettably not assessed after 1.2 °C of warming in any species.

**Statistical analysis**

*Assessing initial thermal limits.* We used mean CTmax as our estimate of upper CTL, as it is often used to estimate vulnerability[5,6]. For egg-to-adult viability and male/female fertility we calculated both LT50 and LT80 thresholds of viability/sterility to estimate upper thermal limits for these traits. These thresholds were selected because we have little knowledge on what threshold is important for population persistence. The LT50 threshold would be most comparable to mean CTmax, while the LT80 threshold represents a more significant loss of fertility/viability that is likely to severely threaten population persistence. We also chose to calculate both thresholds, as LT50 might not be equivalent for sterility versus viability because viability at benign temperatures is rarely 100% (and is often closer to 80%), while fertility is closer to 100% at benign temperatures.

To estimate LT50 and LT80 for male and female sterility and egg-to-adult viability, we used a dose-response model using the *drc* package (version 3.0-1)[65] in R (version 3.6.3). Given that fertility and survival decreased to zero at higher developmental temperatures, we fitted a three-parameter log-logistic dose-response model, which fixes the lower limit to zero. We then estimated the effective dose (LT50 and LT80), which is the temperature resulting in a 50 and 80% reduction in the average response relative to the upper and lower limits of the mean model function.

*Associations between traits and extinction/decline.* To examine whether species differed in the temperature at which they went extinct, and whether extinction temperature differed across widespread and tropical species, we used a general linear model (using the *lme4* package (version 1.2-21[66] in R)), with species and

distribution designated as fixed effects and species nested within distribution. The significance of these effects was then tested using analysis of variance (ANOVA Type II, *car* package (version 3.0-6)[67] in R).

We used linear regressions (using the *lme4* package[66] in R) to examine the relationship between male and female $FTL_{50}/FTL_{80}$, $VTL_{50}/VTL_{80}$, CTmax and *T*opt and mean extinction temperature across the six species exposed to experimental warming. We also looked at the association between these traits and the mean temperature when the estimated population size of the selected lines was significantly lower than ($P < 0.05$, two-sided, independent samples *t*-test) control lines (mean decline temperature) and when there were no longer pupae present (mean sterility temperature), signifying a loss of reproductive capacity. A generalised linear model (using the *lme4* (version 1.2–21[66]) in R), with replicate line (random) nested within treatment (fixed), was used to test for a significant difference between the population size of control and selected lines at each generation. To explore whether different thermal traits were correlated, we conducted multiple pair-wise Pearson's correlations in *R*.

*Associations between traits and environment.* We used linear regressions[66] to examine the relationship between male upper FTLs, VTLs, mean CTmax and *T*opt and climatic variables across all ten species. Given that only $VTL_{80}$ showed a significant association with extinction temperature and male $FTL_{50}$ showed stronger associations with extinction temperature than male $FTL_{80}$ (Supplementary Fig. S4), we chose to focus on male $FTL_{50}$ and $VTL_{80}$. This difference between $VTL_{50}$ and $VTL_{80}$ was likely driven by the fact that viability was below 100%—even at cooler temperatures—and the longer tail in viability decline with increasing developmental temperature in some species (e.g., *D. buzzatii* and *D. hydei*) (Supplementary Fig. S3).

Temperature predictors included average latitude (commonly used as a proxy for temperature) in Australia, AMT, maximum temperature of the warmest month ($T_{MAX}$) and mean temperature of the warmest quarter ($T_{WARM}$). Because heat tolerance has also been linked to precipitation[6], we also included precipitation variables (annual precipitation ($P_{ANN}$), and precipitation of the wettest quarter ($P_{WET}$)). Distribution data were extracted for each *Drosophila* species from the taxodros website https://www.taxodros.uzh.ch/) and then the environmental data for each distribution data point were extracted from the WorldClim dataset (https://www.worldclim.org) and averaged across the entire species distribution to give a single data point per environmental variable per species. The association between male $FTL_{50}$, $VTL_{80}$, mean CTmax and *T*opt and the predictor variables $T_{MAX}$, $T_{WARM}$, $P_{ANN}$ and $P_{WET}$ were also compared using a multiple regression approach[6,66].

*Thermal safety margins and warming tolerance.* Since upper CTLs using CTmax estimate acute heat tolerance across hours of exposure, while FTLs and *T*opt were estimated during development (days of exposure), we calculated both warming tolerance and thermal safety margins for CTmax, male $FTL_{50}$ and *T*opt across the ten species. Warming tolerances for each trait/species were estimated as the difference between $T_{MAX}$ and mean CTmax, male $FTL_{50}$ and *T*opt for each species[6], while thermal safety margins were calculated as the difference between $T_{WARM}$ and mean CTmax, male $FTL_{50}$ and *T*opt for each species[5]. We then used linear regressions[66] to examine the relationship between warming tolerance/thermal safety margins and latitude.

*Selection assessments.* To analyse differences in male fertility/sterility and CTmax between selection and control lines, we used generalised linear mixed models using the glmer function in the *lme4* package[66] in R. Sterility was modelled using a binomial distribution, using the logit link function, while fertility was modelled using a Gaussian distribution. Visual inspection of model diagnostic plots (using the *DHARMa* package (version 0.2.7)[68] in R) showed that the assumptions of parametric analyses were fulfilled. For all traits, 'treatment' (control or selection) and 'temperature' (control or selected developmental temperature) were treated as fixed effects, and 'replicate line' was nested within treatment as a random effect. The significance of fixed effects was then tested using analysis of variance (ANOVA Type III, *car* package (version 3.0-6)[67] in R) and a post hoc test was used to test for significant differences between treatments and temperature (*lsmeans* package (version 2.30-0)[69] in R). Semi-partial $R^2$ estimates[70] were calculated using the *r2glmm* package (version 0.1.2) in R. Separate analyses were performed at each generation and for each species.

**Reporting summary.** Further information on research design is available in the Nature Research Reporting Summary linked to this article.

## Data availability
The authors declare that the data supporting the findings of this study are available within the paper and its supplementary information files. Distribution data were extracted for each *Drosophila* species from the taxodros website (https://www.taxodros.uzh.ch/) and environmental data for each distribution data point were extracted from the WorldClim dataset (https://www.worldclim.org). Source data are provided with this paper.

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

## Acknowledgements

We are grateful to Tamblyn Coadie, Taylor Graham, Vanessa Kellermann, Fiona Cockerall, Winston Ye, Clementine Lasne, Avi Chakraborty, Brooke Zanco and Teresa Kutz for technical support and Jon Bridle and Eleanor O'Brien for field flies. We thank the Australian Research Council for funding B. V. H. and C. M. S. via their fellowship and discovery schemes (DE150100507, DP180103725) and Monash University for their support via their Women in Science and Advancing Diversity schemes.

## Author contributions

B. V. H. conceived and designed the experiment with input from C. M. S. B. V. H. performed the experiments and analysed the data, with input from C. M. S. C. M. S. provided equipment for the experiments. B. V. H. wrote the paper with input from C. M. S. Both authors contributed to the interpretation of the results and approved the final manuscript.

## Competing interests

The authors declare no competing interests.
