## [Peer Review File · Nature Communications]

Reviewers' Comments:

Reviewer #1:

Remarks to the Author:

In the manuscript, "Male fertility thermal limits drive vulnerability to climate warming", the authors work to demonstrate the relationship between various thermally driven life history traits and abiotic climate factors. Specifically, they evaluate the correlations between developmental temperature fluctuations on life history. Then they relate this to environmental weather data. The authors conclude that, while upper thermal limits are commonly used, the fertility thermal limit in males is lower and more tightly correlated to current distributions and thus more relevant to evaluate in the context of global change. Using experimental evolution, they attempt to simulate a fast-paced warming scenario and quantify the evolvability of three different thermal limits. They conclude (like others with respect to upper thermal limits) that there is little potential for populations to be saved through plasticity or evolution.

This is an impressive study that was a huge amount of work. The authors used a combination of 3 temperature fluctuations and measures over experimental evolution to understand how acute plasticity and rapid evolution combine to drive population responses to climate change. However, the nature of this project means that the environmental inference is based on correlation alone. To this end, I am not sure you can say something is "driving" natural populations.

Further, this with extinction temperature. It is an interesting idea but I feel the authors needs to be more upfront with how unrealistic this is (especially in a seasonal environment). There is some mention of seasonality in the discussion but perhaps some more interpretation about what this number actually means. There is also the matter of behavioral regulation which is briefly mentioned as well. The extinction temperature is, in a way, very interesting. But, especially for an audience that does not think about thermal biology all the time, I think the manuscript would benefit from some further interpretation.

Specific comments:

In the abstract and elsewhere, you refer CTLs when you really mean upper thermal limits. As far as I can read, you do not really consider/test/discuss lower thermal limits. I believe it will improve the readability if you are clearer about this.

Throughout the manuscript, the writing heavily relies on "-" in the place of commas, semi-colons or full stops. This makes the paper somewhat difficult to read. Please consider revising

Line 24-26: very difficult to follow this thought.

I understand that space is short but it is a bit difficult to have to look up a supplementary table to get an overview of the species you use. Is this something that could be included in a figure (maybe even an overview figure)?

You are using literature values for T_{opt} but it is important to note these were measured under constant temperature regimes.

Line 171: You are missing a citation

Table S6: should be included in main results?

Line 224 onwards: If animals are using behavioral thermoregulation, how then should the reader interpret your results? What does this mean for your "extinction temperature"?

In the experimental evolution experiment, the pace is exceptionally fast. Further, you say 14 days is, on average, a generation but that is just not true for all of the species you are using. Is there somewhere you can list the average generation times or could you state when this is not the case? I would be very curious to see if those species with longer generation times also scored lower extinction temperatures.

As far as I can read, the statistical analyses all seem appropriate. I would like to encourage the authors to include a table of effect sizes or variance partitioning for the linear models as it will assist in the reader.

Figures 4 and 5 are not the nicest figures. Is there any other way you can visualize this?

Reviewer #2:

Remarks to the Author:

In this study, van Heerwaarden and Sgrò use a generously replicated and carefully controlled set of experiments to reveal some extremely important discoveries about thermal tolerance and evolutionary potential in the face of climate change across a representative set of *Drosophila* fruit fly species.

This is one of the finest manuscripts I have read and reviewed in a very long time, both in presentation and content, so many thanks and high congratulations to the two authors for such outstanding work. The writing is clear, the graphics and visuals equally so, and the discoveries are original, profound and have very high impact.

The study significantly advances our understanding of how biodiversity will be impacted by climate change. It's important that we study these impacts in ectothermic models, which comprise the majority of biodiversity, and we know very little about how species and populations are specifically impacted by thermal change.

This study provides us with not one, but FOUR very important discoveries:

- 1) Tropical species exist much closer to their thermal limits than more widespread species, so are more vulnerable to climate change.
- 2) It is the upper limit of male fertility which determine a species' overall thermal limits, and male fertility's upper thermal threshold always sits below the equivalent female reproductive limit across a number of species. Both reproductive limits sit well below the lethal limit for every species.
- 3) Male thermal fertility limits explain the current geographic distribution across 10 different *Drosophila* species, cementing the importance of thermal constraints on (male) reproduction.
- 4) Upper thermal limits have very little adaptive potential, either phenotypically or genotypically, providing a most concerning finding for future biodiversity under climate change.

These are very very important discoveries, and I really like the way that they can be so assertively presented in the manuscript, without the usual ambiguities or caveats, because they are founded on rigorous and representative experimental work.

The methods are very well described and executed, and the product of a heroic amount of lab work to generate healthy sample sizes given the breadth of the work. I think the approach to gradually raise temperatures for populations through multi-generational time, and then measure how different

species succumb or adapt is extremely revealing, and is an excellent design for understanding how gradual but increasing climate change (albeit accelerated in the lab) will impact on populations.

A range of responsive traits are measured and compared, and the results are clear and logical, but also very important. We know that biodiversity is responding to climate change, but there is a dearth of research on what specific mechanisms or traits are especially sensitive to thermal change, and the nature and extent of their adaptive potential. Here, using comparative studies across six to ten fruit fly model species, the findings are beautifully clear: male reproductive potential is a critically sensitive trait, there is limited potential for this trait to adapt, and tropical species lie closer to their limits and are therefore more vulnerable to unnatural change.

I can barely find anything to improve the study, which is unusual but very nice. Yes, we could think about alternative ways to change / vary / apply temperature through the experiments, but I think the authors have taken the best and most sensible all-in approach for studying how climate change will impact, and the early exposures are relevant to nature. My only suggestion is that the fertility assays could possibly even underestimate the thermal impact on reproductive fitness, because the authors conservatively only measured whether an individual was sterile or had some fertility. We have found in our *Tribolium* model (and using a different 'extreme heat' approach) that subfertility as well as complete sterility can occur across individual males following exposure to heatwave-like conditions, so a brief mention of the contribution of subfertility as well as complete sterility to population declines might add to the impact?

Other than that, congratulations again on a truly fantastic study which will make a major contribution to the very important understanding of how climate change will impact on biodiversity.

REVIEWER COMMENTS

Reviewer #1 (Remarks to the Author):

In the manuscript, “Male fertility thermal limits drive vulnerability to climate warming”, the authors work to demonstrate the relationship between various thermally driven life history traits and abiotic climate factors. Specifically, they evaluate the correlations between developmental temperature fluctuations on life history. Then they relate this to environmental weather data. The authors conclude that, while upper thermal limits are commonly used, the fertility thermal limit in males is lower and more tightly correlated to current distributions and thus more relevant to evaluate in the context of global change. Using experimental evolution, they attempt to simulate a fast-paced warming scenario and quantify the evolvability of three different thermal limits. They conclude (like others with respect to upper thermal limits) that there is little potential for populations to be saved through plasticity or evolution.

Comment: This is an impressive study that was a huge amount of work. The authors used a combination of 3 temperature fluctuations and measures over experimental evolution to understand how acute plasticity and rapid evolution combine to drive population responses to climate change. However, the nature of this project means that the environmental inference is based on correlation alone. To this end, I am not sure you can say something is “driving” natural populations.

Response: We agree with the reviewer that experiments in the laboratory do not have the power to directly assess extinction vulnerability and its drivers in natural populations. Nonetheless, we believe that this work provides strong evidence that male fertility at high developmental temperatures (reflecting average air temperature fluctuations in nature) is closely aligned with extinction vulnerability in the laboratory and shows stronger associations with climatic variables in nature than other measures of thermal tolerance. We are now careful to reiterate that these extinction temperatures are specific to the laboratory (highlighted in resubmitted ms). We have also changed the title from “Male fertility thermal limits *drive* vulnerability to climate warming” to “Male fertility thermal limits *predict* vulnerability to climate warming” and a sentence in the conclusions that male FTLS are “a strong predictor of extinction temperature in the laboratory.”, rather than “driving” extinction vulnerability.

Comment: Further, this with extinction temperature. It is an interesting idea but I feel the authors needs to be more upfront with how unrealistic this is (especially in a seasonal environment). There is some mention of seasonality in the discussion but perhaps some more interpretation about what this number actually means. There is also the matter of behavioral regulation which is briefly mentioned as well. The extinction temperature is, in a way, very interesting. But, especially for an audience that does not think about thermal biology all the time, I think the manuscript would benefit from some further interpretation.

Response: We have now added a sentence in the discussion of the adaptation results “By using a fluctuating temperature regime based on average summer air temperatures and increasing the temperature every generation (see methods), the impacts of microclimate and seasonal temperature variation were not considered. Consequently, the level of warming may have been more rapid or intense than in

nature, where the exploitation of cooler microhabitats through behavioural thermal regulation, or recovery from heat injury during cooler months are possible”

We have also added an extra paragraph in the methods “Although this accelerated warming scenario is not entirely ecologically realistic as it does not encompass seasonal (where reprieve from hotter summer temperatures will occur during cooler months) or microclimate (which may allow species the opportunity to use behavioural thermoregulation to avoid thermal stress) temperature variation, as mentioned above, it is impossible to capture this complexity in the laboratory. Nonetheless, it does allow us to compare extinction risk and adaptation across species in response to equivalent levels of warming in the laboratory. Furthermore, given that successive and extended heat waves are projected in the coming decades, this warming scenario may become increasingly relevant to small, short lived ectotherms like *Drosophila*.”

Specific comments:

In the abstract and elsewhere, you refer CTLs when you really mean upper thermal limits. As far as I can read, you do not really consider/test/discuss lower thermal limits. I believe it will improve the readability if you are clearer about this.

Response: Thanks for pointing out this flaw. We are now clearer that we are referring to upper CTLs and CTmax in the abstract and in the introduction.

Throughout the manuscript, the writing heavily relies on “-“ in the place of commas, semi-colons or full stops. This makes the paper somewhat difficult to read. Please consider revising

Response: We have removed many of the “-“ to make the ms easier to read.

Line 24-26: very difficult to follow this thought.

Response: We have changed this to “Male fertility thermal limits, which are much lower than CTmax, are better predictors of species’ current distributions and extinction in the laboratory.” which we hope is clearer.

I understand that space is short but it is a bit difficult to have to look up a supplementary table to get an overview of the species you use. Is this something that could be included in a figure (maybe even an overview figure)?

Response: Given that we have already added an additional table to the ms (see response to comment below), we feel that it would be too much to include an overview figure. Instead, we have added additional information on the species used in the evolution experiment in the species key on Figure 1.

You are using literature values for T_{opt} but it is important to note these were measured under constant temperature regimes.

Response: We are now clear when we first refer to these estimates that they were estimated using constant temperature regimes (approx. line 113)

Line 171: You are missing a citation

Response: We have added the missing citation

Table S6: should be included in main results?

Response: This is now included in the main results

Line 224 onwards: If animals are using behavioral thermoregulation, how then should the reader interpret your results? What does this mean for your “extinction temperature”?

Response: Here the reviewer is referring to “These findings - in conjunction with results showing this trait does not relate to habitat temperatures [3,6,12,13] - suggest that species may be using behavioural thermal regulation to evade temperatures above their T_{opt} [10], or this trait is not important for driving population persistence”.

Here we have now added “Given that species’ decline/ extinction temperatures were more than 2 °C above their T_{opt} , and male fertility was closely associated with both laboratory extinction and habitat temperatures, it is likely that male fertility, rather than T_{opt} , is a key determinant of population persistence and vulnerability to high temperatures.”

We also further discuss the role of behavioural thermoregulation in the methods and discussion (see response above).

In the experimental evolution experiment, the pace is exceptionally fast. Further, you say 14 days is, on average, a generation but that is just not true for all of the species you are using. Is there somewhere you can list the average generation times or could you state when this is not the case? I would be very curious to see if those species with longer generation times also scored lower extinction temperatures.

Response: The average generation times are in Table S1. We have also added a comment in the second paragraph of the “Male upper fertility thermal limits drive warming extinction” section showing that there is no association between developmental time and extinction temperature... “Given that the species differed in their generation times (Table S1), we also explored whether species with longer development times had lower extinction temperatures. We found no association between average extinction temperature and average development time at 25 (Adjusted $R_2 = 0.27$, $df = 5$, $P = 0.16$) or 28 °C (Adjusted $R_2 = 0.17$, $df = 5$, $P = 0.23$) (Figure S5).”

As far as I can read, the statistical analyses all seem appropriate. I would like to encourage the authors to include a table of effect sizes or variance partitioning for the linear models as it will assist in the reader.

Response: We have now included semi-partial R^2 and estimates (B) for the fixed effects.

Figures 4 and 5 are not the nicest figures. Is there any other way you can visualize this?

Response: We played around with various ways to visualise these results, but as there are so many different lines/ treatments and species, and we wanted to show the fertility results alongside the sterility data, we believe that the original figures are the

clearest. In addition, reviewer 2 was impressed with the quality and clarity of the figures, so we have chosen to keep them as is.

Reviewer #2 (Remarks to the Author):

In this study, van Heerwaarden and Sgrò use a generously replicated and carefully controlled set of experiments to reveal some extremely important discoveries about thermal tolerance and evolutionary potential in the face of climate change across a representative set of *Drosophila* fruit fly species.

This is one of the finest manuscripts I have read and reviewed in a very long time, both in presentation and content, so many thanks and high congratulations to the two authors for such outstanding work. The writing is clear, the graphics and visuals equally so, and the discoveries are original, profound and have very high impact.

The study significantly advances our understanding of how biodiversity will be impacted by climate change. It's important that we study these impacts in ectothermic models, which comprise the majority of biodiversity, and we know very little about how species and populations are specifically impacted by thermal change.

This study provides us with not one, but FOUR very important discoveries:

1) Tropical species exist much closer to their thermal limits than more widespread species, so are more vulnerable to climate change.

2) It is the upper limit of male fertility which determine a species' overall thermal limits, and male fertility's upper thermal threshold always sits below the equivalent female reproductive limit across a number of species. Both reproductive limits sit well below the lethal limit for every species.

3) Male thermal fertility limits explain the current geographic distribution across 10 different *Drosophila* species, cementing the importance of thermal constraints on (male) reproduction.

4) Upper thermal limits have very little adaptive potential, either phenotypically or genotypically, providing a most concerning finding for future biodiversity under climate change.

These are very very important discoveries, and I really like the way that they can be so assertively presented in the manuscript, without the usual ambiguities or caveats, because they are founded on rigorous and representative experimental work.

The methods are very well described and executed, and the product of a heroic amount of lab work to generate healthy sample sizes given the breadth of the work. I think the approach to gradually raise temperatures for populations through multi-generational time, and then measure how different species succumb or adapt is extremely revealing, and is an excellent design for understanding how gradual but increasing climate change (albeit accelerated in the lab) will impact on populations.

A range of responsive traits are measured and compared, and the results are clear and logical, but also very important. We know that biodiversity is responding to climate change, but there is a dearth of research on what specific mechanisms or traits are especially

sensitive to thermal change, and the nature and extent of their adaptive potential. Here, using comparative studies across six to ten fruit fly model species, the findings are beautifully clear: male reproductive potential is a critically sensitive trait, there is limited potential for this trait to adapt, and tropical species lie closer to their limits and are therefore more vulnerable to unnatural change.

I can barely find anything to improve the study, which is unusual but very nice. Yes, we could think about alternative ways to change / vary / apply temperature through the experiments, but I think the authors have taken the best and most sensible all-in approach for studying how climate change will impact, and the early exposures are relevant to nature. My only suggestion is that the fertility assays could possibly even underestimate the thermal impact on reproductive fitness, because the authors conservatively only measured whether an individual was sterile or had some fertility. We have found in our *Tribolium* model (and using a different 'extreme heat' approach) that subfertility as well as complete sterility can occur across individual males following exposure to heatwave-like conditions, so a brief mention of the contribution of subfertility as well as complete sterility to population declines might add to the impact?

Other than that, congratulations again on a truly fantastic study which will make a major contribution to the very important understanding of how climate change will impact on biodiversity.

Response: We thank this reviewer for their interest and support of our research. We have now added several sentences in the “Upper critical thermal limits underestimate vulnerability” section addressing this idea.

“Although we found that male FTL_{50} estimates were strongly linked to extinction in the laboratory, they may potentially underestimate the effects of climate warming on population persistence in nature, as reductions in male fertility (sub-fertility) can occur prior to complete sterility and accumulate across generations [16]. We also did not consider the impact of extreme temperatures, which in combination with above average developmental temperatures, may exacerbate the effects of heat injury on fertility [17]. Future studies assessing both complete and sub-fertility within and across generations, under different thermal regimes, may further elucidate the potential impacts of climate warming on male fertility and population persistence.”

Reviewers' Comments:

Reviewer #1:

Remarks to the Author:

As I said in my original review, this manuscript represents a huge amount of work and very interesting results.

I appreciate that the authors have toned down the use of "drive" in the title and throughout the manuscript. It is very interesting that it may explain some observed natural patterns but it is important to acknowledge the limitations as well. I also appreciate that you have specified upper thermal limits throughout the manuscript.

My other main concern in the original review has to do with the calculation of extinction rate. A lot of the analysis and interpretation hangs on this metric. It is very good that the authors have now acknowledged in the methods and to a lesser extent in the discussion that the selection regime imposed to calculate extinction rate are unrealistic. I do feel the argument in the discussion is a little undercut by the end of the paragraph,

"Nonetheless, given that successive and extended heat waves are projected in the coming decades 328 [1,2], warmer than average temperatures across multiple generations (which may be particularly 329 relevant to small, short lived ectotherms), in conjunction with extreme temperatures, are likely are

330 likely to push many species to extinction. Projected increases in night time temperatures are also 331 likely to further reduce opportunities for fertility recovery and exacerbate effects of high 332 temperatures on fertility [62]."

Please note that here and in my original review, I am not saying it was the wrong approach. I just think you need to be careful about how it is interpreted. I think "extinction rate" sounds like something that could be plugged into a model or otherwise "applied" but, in reality, it is more of a population endurance measure.

Overall, I think this is a solid paper. A very large experiment with good data, well analyzed and interpreted. It is a nice example of a trait that is not often considered being more closely correlated to occurrence than those most commonly used.

Reviewer #1 (Remarks to the Author):

As I said in my original review, this manuscript represents a huge amount of work and very interesting results.

I appreciate that the authors have toned down the use of “drive” in the title and throughout the manuscript. It is very interesting that it may explain some observed natural patterns but it is important to acknowledge the limitations as well. I also appreciate that you have specified upper thermal limits throughout the manuscript.

My other main concern in the original review has to do with the calculation of extinction rate. A lot of the analysis and interpretation hangs on this metric. It is very good that the authors have now acknowledged in the methods and to a lesser extent in the discussion that the selection regime imposed to calculate extinction rate are unrealistic. I do feel the argument in the discussion is a little undercut by the end of the paragraph,

“Nonetheless, given that successive and extended heat waves are projected in the coming decades

328 [1,2], warmer than average temperatures across multiple generations (which may be particularly

329 relevant to small, short lived ectotherms), in conjunction with extreme temperatures, are likely are

330 likely to push many species to extinction. Projected increases in night time temperatures are also

331 likely to further reduce opportunities for fertility recovery and exacerbate effects of high
332 temperatures on fertility [62].”

Please note that here and in my original review, I am not saying it was the wrong approach. I just think you need to be careful about how it is interpreted. I think “extinction rate” sounds like something that could be plugged into a model or otherwise “applied” but, in reality, it is more of a population endurance measure.

Overall, I think this is a solid paper. A very large experiment with good data, well analyzed and interpreted. It is a nice example of a trait that is not often considered being more closely correlated to occurrence than those most commonly used.

Response: We have added an extra sentence (“Consequently, the extinction temperatures estimated in this study may differ to extinction temperatures in nature”) and amended these last two sentences to highlight that we are referring to the effects of these scenarios on fertility (“Nonetheless, given that successive and extended heat waves are projected in the coming decades [1,2], warmer than average temperatures across multiple generations (which may be particularly relevant to small, short lived ectotherms), in conjunction with extreme temperatures, are likely to push many species beyond their fertility limits and elevate extinction risk. Projected increases in night time temperatures may also further reduce opportunities for fertility recovery and exacerbate effects of high temperatures on fertility [62].”).